# On the Fast Convergence of Unstable Reinforcement Learning Problems

## Abstract

For many of the reinforcement learning applications, the system is assumed to be inherently stable and with bounded reward, state and action space. These are key requirements for the optimization convergence of classical reinforcement learning reward function with discount factors. Unfortunately, these assumptions do not hold true for many real world problems such as an unstable linear–quadratic regulator (LQR)[1]. In this work, we propose new methods to stabilize and speed up the convergence of unstable reinforcement learning problems with the policy gradient methods. We provide theoretical insights on the efficiency of our methods. In practice, our method achieve good experimental results over multiple examples where the vanilla methods mostly fail to converge due to system instability.

## 1 Introduction

Reinforcement learning (RL), powered by the generalization ability of machine learning structures, has been fairly successful in classical control tasks (Hafner & Riedmiller, 2011; Lillicrap et al., 2016) and problems like Atari (Mnih et al., 2013) and Go (Silver et al., 2016). RL aims to train a policy to achieve maximum reward or minimize the cost[2], which is similar to control theoretic approaches on designing a controller. However, different from classical optimal control which requires full knowledge of transition dynamics, RL can learn the optimal policy from the past data directly by solving an optimization problem without the knowledge of the underlying dynamics.

One of the mainstream methods to solve an RL problem is policy optimization via gradient descent. However, the convergence of the policy optimization algorithm heavily relies on an unapparent yet critical assumption of the system dynamics itself: stability[3]. In addition, to ensure the convergence of policy optimization, it requires the Lipschitz property of the cost function and its gradient. In fact, in many of the existing RL benchmark examples such as OpenAI's classical control environments, the state space/actions/costs are clipped to ensure that the policy would never move to extreme conditions and costs/states are bounded, in order to reduce the error derivatives (Mnih et al., 2015). Unfortunately, similar formulations are not directly applicable to unstable systems, such as a LQR with an unstable state matrix (i.e. the spectral radius of the state matrix is outside the unit circle), as the standard policy gradient based methods are likely to fail.

Motivated by the above issue, in this paper we aim to enable and speed up the convergence of policy gradient methods for unstable RL problems, we propose a logarithmic mapping method on loss functions supported by rigorous theoretical proofs and experimental results. The key contributions are summarized as follows.

- We formally define the unstable RL problem in the scope of "input-to-output" stability, with the input actions leading to a temporal growing effect against cost output. This is the first

---

[1] By unstable LQR we mean the state transition matrix $A$ in LQR (see Equation (1)) has a spectral norm outside the unit circle.

[2] For the rest part of this paper, we will use "minimizing cost" as objective since this is closely related to optimal control and optimization.

[3] In this paper, "stability" denotes "input-to-output" stability, where a small perturbation of the input signal (control action/state perturbation) will not lead to a large deviation in the system output cost. We use model-free methods in this paper, therefore the system output target is the cost function. For formal stability definition, please refer to Section 3.1

time the convergence issue of unstable RL problems is studied, we demonstrate that a major issue for policy gradient methods on unstable RL problems is the slow convergence rate, which is the due to large spectral radius of the Hessian matrix.

- We propose a simple yet effective logarithmic mapping to alleviate this issue and speed up the convergence. We show both theoretical advantage and experimental results to support the contribution of faster convergence rate. Notably, our finite horizon problem setup does not require the bounded assumption of cost function. The experiments cover LQR examples to customized nonlinear cases with neural network based policy.

- We provide an efficient method to find a better initialization of control policy by optimizing it over the spectral norm of controlled system. We use it as a fast pre-processing step to effectively save the computation cost and allow larger learning rate for fast convergence.

## 2 BACKGROUND AND RELATED WORK

### 2.1 UNSTABLE DYNAMICAL SYSTEM

For many real-world unstable dynamical systems with limited explicit knowledge and practical aspects, system identification (SYSID) (Mettler et al., 1999; Ananth & Chidambaram, 1999; Bond & Daniel, 2008) with control (Jordan & Jacobs, 1990; Arora et al., 2011) is a 2-step approach to maneuver such a system, where the SYSID step targets on learning the system and the "control" step designs a controller to stabilize the system and minimize the cost. Despite most of the existing works focus on the system instability itself, not many works realize the rise of optimization issues due to the diverging nature of the unstable system. Shahab & Doraiswami (2009) pointed out that the aforementioned vanilla approach may fail the system identification due to the rich sampling space required by unstable system and thus they proposed a close-loop algorithm to accommodate prior information into the identification process. Nar et al. (2020) noticed the imbalanced sample influence of unstable systems and used a time-weight loss to alleviate the effect. With a growing popularity of data-driven application in control/RL problems, the optimization issues from unstable systems naturally extend to many trending methods such as policy gradient. To the best of our knowledge, this work is the first attempt to investigate the convergence issues of policy gradient algorithm on unstable systems and propose practical methods to alleviate this issue.

### 2.2 LQR PROBLEM

For a discrete-time linear system, its state equation is represented by:

$$x_{t+1} = Ax_t + Bu_t \tag{1}$$

where, $x_t \in \mathbb{R}^n$ and $u_t \in \mathbb{R}^m$ denote the system state and control action at time step $t$, $A \in \mathbb{R}^{n \times n}$ and $B \in \mathbb{R}^{n \times m}$ are the system transition matrices. The feedback gain is parameterized by the matrix $K \in \mathbb{R}^{m \times n}$ with

$$u_t = -Kx_t. \tag{2}$$

The intermediate cost function is in the quadratic form of state $x_t$ and control $u_t$, where $Q \in \mathbb{R}^{n \times n}$ and $R \in \mathbb{R}^{m \times m}$ are given positive definite matrices to parameterize the quadratic cost. The optimal control problem can be formulated as minimizing the following target over $K$:

$$C_{K,T} \triangleq \mathbb{E}_{x_0 \sim D} \left[ \sum_{t=0}^{T} x_t^\top Q x_t + u_t^\top R u_t \right] \tag{3}$$

where $x_t = (A - BK)^t x_0$ by plugging in $u_t = -Kx_t$ and $D$ is the distribution of initial state. If $T \to \infty$, then the problem is called *infinite* horizon LQR, otherwise when $T$ is a finite positive integer, it is called *finite* horizon LQR.

According to Nise (2020), a $T$-time-step system is controllable if we can reach any target state $x^*$ from any initial state $x_0$. The necessary and sufficient condition for controllability is that the controllability gramian is full rank, i.e.,

$$\text{rank}([A^{T-1}B, A^{T-2}B, \cdots, AB, B]) = n, \tag{4}$$

where $n$ is the dimension of the state. In this paper, we assume the LQR is controllable. For unstable LQR in the infinite horizon, the cost function is not traceable since $C_{K,T} \to \infty$ when $T \to \infty$. To bypass this barrier, in this paper we focus on a finite horizon case and consider the cost for the first $T$ steps, which is consistent with our unstable RL setup later. In practice, infinite horizon cases are similarly approximated by finite step trajectories (Fazel et al., 2018) so that the implementations are identical. To distinguish the finite horizon LQR setup in Hambly et al. (2021), we have the policy matrix $K$ fixed along the trajectory instead of a time-varying $\{K_t\}$ for $t = 0, 1, ..., T$.

## 2.3 CONVERGENCE OF POLICY GRADIENT METHODS

Despite the increasing popularity of policy gradient method, there has been little understanding on the convergence property of such methods. Only until recently, there are some work trying to tackle this problem from different perspectives. Zhang et al. (2019) proved the asymptotic convergence of policy gradient methods to stationary points, and later they extended the work and proposed MRPG (Modified Random-Horizon Policy Gradient) to almost surely converge to actual local optimal policies (Zhang et al., 2020). Bhandari & Russo (2019) studied the convergence rate for policy gradient methods under gradient dominance condition and they attained a linear convergence rate in Bhandari & Russo (2021). Agarwal et al. (2020) extensively studied the convergence rate under various combination of algorithms and function approximation for both first-order and quasi second-order policy gradient methods.

As a special setting of RL, there is a rising interest in studying the convergence of policy gradient method under LQR application. Fazel et al. (2018) was the first to achieve the global convergence of policy gradient methods for infinite-horizon LQR problems, by proving the gradient domination and smoothness conditions. In their work, the infinite-horizon setup and smoothness/bounded cost assumptions are dependent on system stability, these assumptions no longer holds if $A - BK$ becomes unstable and objective function goes to infinity. In practice, the vanilla policy gradient method struggles to converge if the autonomous system is unstable, as we will also show in our experiments in Section 4.1. Bhandari & Russo (2019) extends the LQR setup to a more general class of control policies and identifies the structural conditions for convergence to optimal stationary point. Perdomo et al. (2021) proposes to stabilize a dynamical system with a discounted annealing algorithm by gradually increasing discount factor $\gamma$ dependent on the system and current policy, which can ensure the boundedness of infinite-horizon system cost. In the scenario of finite horizon LQR and stochastic noise, Hambly et al. (2021) provides a global linear convergence guarantee. Tu & Recht (2018) studied Least-Squares Temporal Difference (LSTD) method on LQR and number of samples needed for LSTD estimator of value function.

Nevertheless, all the above work require the assumption of the system to be stable under the policy throughout optimization, or equivalently, $A - BK$ has a spectral radius less than 1. Unfortunately, with an unstable $A$ and random initialization of policy $K$, this assumption is mostly invalid. In contrast, our proposed algorithm target at a finite horizon setup which shares an identical implementation with infinite horizon based methods but does not require this assumption at all.

## 3 PROPOSED METHODS

### 3.1 FORMULATING INSTABILITY

In the dynamical system literature, stability usually denotes input-to-state stable (ISS) (Sontag & Wang, 1995) in system dynamics, where a small deviation of system state or control action perturbation will not lead to dramatic change of future states. Formally, consider a general continuous dynamical system $\dot{x} = f(x, u)$ with continuously differentiable $f(\cdot)$ and $x(t, x_0, u)$ denote the trajectory of $x$ given initial condition $x_0$ and control feedback $u$. Then the system is ISS if there exist $\mathcal{K}$ function $\gamma : \mathbb{R}^+ \to \mathbb{R}^+$ and $\mathcal{KL}$ function $\beta : \mathbb{R}^+ \times \mathbb{R}^+ \to \mathbb{R}^+$, s.t.

$$\textbf{ISS:} \qquad \|x(t, x_0, u)\| \leq \gamma(\|u\|_\infty) + \beta(\|x_0\|, t). \qquad (5)$$

where $\|u\|_\infty = sup\{\|u(t)\|\} < \infty$ for $t \geq 0$. Function $\gamma(\cdot)$ is called $\mathcal{K}$ function if $\gamma(\cdot)$ is continuously increasing and $\gamma(0) = 0$, $\beta(\cdot, \cdot)$ is called $\mathcal{KL}$ function if $\beta(\cdot, t)$ is $\mathcal{K}$ function for all the $t \geq 0$.

In this paper, we refer to stability as the input-to-output (I/O) stability (Sontag & Wang, 1999). Let the output $y = h(x(t))$ be a function of $x$, for instance, $y$ can be the quadratic function to regulate the error or other system properties we wish to stabilize. The system is input-to-output (I/O) stable if there exist $\mathcal{K}$ function $\gamma(\cdot) : \mathbb{R}^+ \to \mathbb{R}^+$ and $\mathcal{KL}$ function $\beta(\cdot, \cdot) : \mathbb{R}^+ \times \mathbb{R}^+ \to \mathbb{R}^+$, s.t.

$$\textbf{I/O stable:} \qquad \|y(x(t, x_0, u))\| \leq \gamma(\|u\|_\infty) + \beta(\|x_0\|, t), \qquad (6)$$

Both ISS and I/O stability indicate bounded inputs leading to bounded system behavior, while there does not exist any causal relationship between the two with arbitrary choice of output function $y(\cdot)$. For further clarification, we provide a linear system example in Appendix A. In a reinforcement learning setting, the "cost-to-go" but not necessarily the whole trajectory is always observable to the agent, because the returned cost serves as the evaluation metric for RL algorithm. Thus, I/O stability is a more suitable stability concept for analyzing general RL problems.

## 3.2 UNSTABLE RL PROBLEMS

**Road-map for convergence rate analysis:**
In this following, we formulate the unstable RL problems and provide a theoretical view of convergence rate bound under the vanilla setting and our proposed method. First, we formulate a Markov decision process (MDP) problem with cost formulation in **Assumption 3.2**, where the cost function is allowed to exponentially grow against time $t$ with some base numbers $\{\phi_k\}$, corresponding to the base number $C$ in **Equation (7)**. We also assume the Lipschitz property (**Assumption 3.4**) and local strong convexity (**Assumption 3.5**) of such $\phi$ against model parameter $\theta$. Based on the above assumptions, we derive the bound of the spectral norm for the Jacobian matrix of the cost function towards parameters (**Theorem 3.9**). For the optimization process, we view it as a dynamical system, translating the necessary condition for monotonic decreasing as bounding the updating step by the inverse of the spectral norm. In order to satisfy such condition for convergence, the learning rate should be bounded and therefore the convergence rate is limited. The deterministic optimization results are summarized in **Theorem 3.12** for vanilla policy gradient and in **Theorem 3.15** for our proposed logarithmic mapping. The stochastic convergence results are discussed in Section 3.3. We defer most of the proofs to **Appendix E** due to space constraints.

### 3.2.1 PROBLEM FORMULATION AND ASSUMPTIONS

To formulate the problem, consider a discrete-time continuous state MDP $\langle \mathcal{S}, \mathcal{A}, \mathcal{P}, \mathcal{C} \rangle$, where $\mathcal{S}$ is the continuous state space, $\mathcal{A}$ is the continuous action space, $\mathcal{P}(s_{t+1}|s_t, a)$ is the transition probability, $c_t(s, a)$ is the immediate cost at time step $t$ and $s_0$ is the initial condition. Assume that the cost is upper bounded by a polynomial of time step, s.t.,

$$|c_t(s, a)| \leq DC^t. \qquad (7)$$

with positive constants $D > 0$ and $C > 0$. The target is to find an optimal policy to minimize the accumulated cost. When $C \leq 1$, the cost is bounded by $D$ independent of $t$ and the system is I/O stable with bounded output being the common setup for RL problems. In this work, we consider a more general setting without the upper bound on $C$. That is, the step cost could grow exponentially with time step $t$, potentially violating Equation (6) where $\gamma(\|u\|_\infty)$ being independent of $t$.

*Remark* 3.1. One way to solve the infinite horizon version of this problem is introducing a discount factor $\gamma$ strictly smaller than $C$. Then the problem could be formulated into classical bounded RL problem with finite expected cost return for infinite steps. However, if $C$ is much larger than 1, this approach does not work because the weights for future time-steps vanishes quickly. A corresponding experiment is shown in a later section (Section 4.1).

Instead, we formulate the problem into finite time horizon of finite time step $T \in \mathbb{Z}^+$, with trajectory cost $v_T(s, \theta)$ as the value function of initial state $s \in \mathbb{R}^n$ and policy parameter $\theta \in \mathbb{R}^d$.

$$v_T(s, \theta) = \mathbb{E}_{s_{t+1} \sim p(s_t, a_t), a_t \sim \pi(s_t, \theta)} \left[ \sum_{t=0}^{T} (c_t | s_0 = s) \right],$$

$$V_T(\theta) = \mathbb{E}_{s \sim \mathcal{D}} \left[ v_T(s, \theta) \right].$$

where $V_T(\theta)$ is the expectation of $v_T(s, \theta)$ over the initial state distribution and the optimization object. For a vanilla policy gradient method, we have update step:

$$\theta \leftarrow \theta - \eta \nabla_\theta V_T(\theta). \tag{8}$$

The gradient has Hessian matrix:

$$J_T(\theta) = \nabla_\theta^2 V_T(\theta). \tag{9}$$

Denote $\rho_{max}(A) = \max\{|\lambda| : \lambda \text{ is an eigenvalue of } A\}$ as the spectral radius of the state matrix $A$, $\rho_{min}(A) = \min\{|\lambda| : \lambda \text{ is an eigenvalue of } A\}$ as the smallest absolute value of eigenvalues.

**Assumption 3.2.** Assume $V_T(\theta)$ can be parameterized by basis function $\mathbb{E}[c_t] \sim \sum_{k=1}^m d_k \phi_k(\theta)^t$, $V_T(\theta) \sim \sum_{k=1}^m d_k \sum_{t=0}^T \phi_k(\theta)^t$, where $d_k > 0$ and $0 \leq \underline{C} < \phi_k(\theta) \leq \overline{C}$.

*Remark* 3.3. The motivation of such basis functions is inspired by the departing output trajectories of unstable systems. Finite horizon LQR example is a special case of our formulation above, where the control matrix $K$ and cost function $C_{K,T}$ corresponds to the policy parameter $\theta$ and cost function $V_T(\theta)$, The LQR step cost $x_t^\top Q x_t + u_t^\top R u_t$ can be rewritten as $x_0^\top (A - BK)^{t\top} (Q + K^\top R K)(A - BK)^t x_0$, which is bounded by $\|x_0\|^2 \|Q + K^\top R K\| \|A - BK\|^{2t}$. The $\|A - BK\|^2$ term corresponds to $\phi$ in Assumption 3.2 with exponential growth with time step $t$ and other terms remain positive constant. Therefore, the finite horizon LQR cost can be formulated into Assumption 3.2 regardless of stability. The formulation is also valid for optimal control problems with polynomial cost functions or other unstable RL problems with exponentially growing cost as a function of time.

**Assumption 3.4.** Assume $\phi_k(\theta)$ is twice differentiable and Lipschitz, the gradient of $\phi_k(\theta)$ is also Lipschitz continuous, s.t.,

$$\|\nabla \phi_k(\theta_1) - \nabla \phi_k(\theta_2)\| \leq L_1 \|\theta_1 - \theta_2\|,$$
$$\|\phi_k(\theta_1) - \phi_k(\theta_2)\| \leq L_2 \|\theta_1 - \theta_2\|.$$

for $L_1, L_2 \in \mathbb{R}^+, \theta_1, \theta_2 \in \mathbb{R}^d$.

**Assumption 3.5.** Assume local $\alpha$-strong convexity of $\phi_k(\theta)$:

$$\phi_k(\theta_1 + \theta_2) \geq \phi_k(\theta_1) + \theta_2^\top \nabla_\theta \phi_k(\theta_1) + \frac{\alpha}{2} \|\theta_2\|^2.$$

for all the $\theta_1, \theta_2 \in \mathbb{R}^d$ and $\theta_* = \underset{\theta \in \mathbb{R}^d}{\arg \min} \, \phi_k(\theta)$

*Remark* 3.6. The strong convexity assumption is indeed a "strong" one for many of the unstable RL examples and policy basis functions. Even for the LQR problems, the cost function is not global-convex (Fazel et al., 2018). The purpose of such assumption is to pave the path for convergence rate analysis. Here we argue that the initialization or the beginning epochs could lead the optimization into a local strong convexity, after which our convergence rate analysis is applicable. Like many other popular optimization methods in ML field such as gradient descent, we do not guarantee the algorithm's convergence to the global optimal. In practice, the stochastic gradient method could find a near-optimal result.

### 3.2.2 CONVERGENCE RATE DERIVATION

**Lemma 3.7.** *Update the value function $V_T(\theta)$ by policy gradient method with $\theta \leftarrow \theta - \eta \nabla_\theta V_T(\theta)$, choose step size $\eta < 2/\underset{\xi \in [0,1]}{\max} \rho_{max}(J_T(\theta - \xi \eta \nabla J_T(\theta)))$, then $V_T(\theta)$ is monotonically decreasing.*

*Remark* 3.8. The idea of Lemma 3.7 is inspired by *Proposition 3.4* in Bertsekas & Tsitsiklis (1996). With the quadratic approximation of the value function, the proper step size is a necessary condition for monotonically increasing because large steps are likely to overshoot. For a multi-dimension problem, the upper bound of the learning rate is dependent on the "steepest" component corresponding to the spectral radius of the Jacobian matrix or the "smoothness" of the function. Different from the well-known convergence result of constant step size optimization for the global L-smooth function, we use the local smoothness between the two steps. In other words, the function is guaranteed to decrease if the step size $\eta$ is small enough to ensure that: (a) the update step is small; and, (b) the $\rho_{max}$ does not change too much.

**Theorem 3.9.** *If $V_T(\theta)$ satisfies Assumption 3.2 and Assumption 3.4, using the vanilla gradient descent algorithm from Equation* (8)*, then $\rho_{max}(J_T(\theta)) < \sum_{k=1}^{m} d_k[L_1(\sum_{t=0}^{T} t\phi_k(\theta)^{t-1}) + L_2^2(\sum_{t=0}^{T} t(t-1)\phi_k(\theta)^{t-2}]$*

**Theorem 3.10.** *Suppose $V_T(\theta)$ satisfies Assumption 3.2 and Assumption 3.4, using the vanilla gradient descent algorithm from Equation* (8)*, if $\eta < 1/\sum_{k=1}^{m} d_k[L_1(\sum_{t=0}^{T} t\phi_k(\theta)^{t-1}) + L_2^2(\sum_{t=0}^{T} t(t-1)\phi_k(\theta)^{t-2}]$, then the update step requirement in Lemma 3.7 for monotonic decrease of value function is satisfied.*

By Theorem 3.10, we claim that to stabilize the convergence for the finite horizon unstable problem, the learning rate $\eta$ needs to be smaller than the inverse of polynomial term of $\max(\phi_k(\theta))$, otherwise the optimization is likely to diverge. At the beginning of optimization, when the system is not well controlled by the feedback policy, a small learning rate is needed. If a fix or decaying learning rate scheduler is used, this constraint requiring a small starting learning rate slows down the policy convergence in the later part of the optimization. Therefore, a customized learning rate which is dependent on the system "instability" is needed for fast and steady convergence.

**Proposition 3.11.** *For twice-differentiable $f$, $f$ is $\alpha$-strong convexity function if and only if $\nabla^2 f(x) \succcurlyeq \alpha I$ for some $\alpha > 0$ and $x \in \mathbb{R}^d$.*

*Proof.* For a twice-differentiable function, $\alpha$-strong convexity is equivalent to the smallest eigenvalue of Hessian of $f$ being lower bounded by $\alpha$. $\square$

**Theorem 3.12.** *Assume $\phi_k(\theta)$ is local strong convex as stated in Assumption 3.5 and the decreasing learning rate $\eta$ satisfies the conditions in Theorem 3.10, then if we run gradient descent for $V_T(\theta)$, it yields a solution:*

$$\|\theta_l - \theta_*\|^2 \leq q^l \|\theta_0 - \theta_*\|^2, \tag{10}$$

*where $\sqrt{q}$ denotes the convergence rate and its square $q$ is lower bounded, s.t., $q \geq (1 - \frac{2\omega^*\alpha}{\rho_{max}(J_T(\theta_0))})$,*

*where $\omega^* = \min_{\theta} \sum_{k=0}^{m} d_k[(\sum_{t=0}^{T} t\phi_k(\theta)^{t-1})]$.*

LOGARITHMIC MAPPING OF FINITE HORIZON VALUE FUNCTION:

In the vanilla setup, the value function of gradient descent is $\widetilde{V}_T(\theta)$. We propose a logarithmic mapping,

$$\widetilde{V}_T(\theta) := log(V_T(\theta)), \tag{11}$$

to regularize the spectral radius of gradient Jacobian and gradient variance. The sampled gradient approximation has the form of

$$\widehat{\widetilde{V}}_T(\theta) := \frac{1}{b} \sum_{j=1}^{b} log(v_T(s_j, \theta)). \tag{12}$$

*Remark* 3.13. Notice the unbiased estimation is in the form of $log(\frac{1}{b}\sum v_T(s_j, \theta))$ instead of Equation (12). We apply logarithm mapping in each realization to reduce the stochastic variance for optimization consideration. Equation (12) can also be viewed as expectation of the stochastic realizations of $log(V_T(\theta))$.

**Theorem 3.14.** *Consider the parameterization of $V_T(\theta)$ and Lipschitz condition in Assumption 3.2 and Assumption 3.4, if we run gradient descent for logarithm mapped $V_T(\theta)$, if $\eta < V_T(\theta)/\sum_{k=1}^{m} d_k[L_1(\sum_{t=0}^{T} t\phi_k(\theta)^{t-1}) + L_2^2(\sum_{t=0}^{T} t(t-1)\phi_k(\theta)^{t-2}]$, then the update step requirement in Lemma 3.7 for monotonic decrease of value function is satisfied.*

*Proof.* The update step under logarithmic mapping is,

$$\theta \leftarrow \theta - \eta \nabla_\theta \widetilde{V}_T(\theta) = \theta - \frac{\eta}{V_T(\theta)} \nabla_\theta V_T(\theta). \tag{13}$$

The learning rate $\eta$ is equivalently normalized by $V_T(\theta)$ and by utilizing Theorem 3.10 we have the proof completed. $\qquad\square$

**Theorem 3.15.** *Assume $\phi_k(\theta)$ is local strong convex as stated in Assumption 3.5 and the decreasing learning rate $\eta_l < \frac{C}{L_1 T + L_2{}^2 T(T-1)}$, then running the gradient descent for logarithm mapped $V_T(\theta)$ with Equation (13) yields a solution*

$$\|\theta_{l+1} - \theta_*\|^2 \le q_l \|\theta_l - \theta_*\|^2,$$

*where the square of the step convergence rate $q_l$ has a varying lower bound s.t. $q_l \ge (1 - \frac{2\omega^*\alpha}{\rho_{max}(J_T(\theta_l))})$.*

*Remark* 3.16. Compared with Equation (15), the $\frac{2\omega^*\alpha}{\rho_{max}(J_T(\theta))}$ term is proportional to the inverse of current spectral radius instead of the maximum spectral radius throughout the optimization trajectory. Considering an unstable system initialized with random policy, the initial $\rho_{max}(J_T(\theta_0))$ could be much larger than $\rho_{max}(J_T(\theta_l))$ for $\theta_l$ in the later part of the optimization. Practically, using the logarithmic mapping achieves a much faster convergence rate.

## 3.3 STOCHASTIC CASE

The above convergence rate analysis is based on deterministic optimization. The convergence of stochastic sampled gradient method with a diminishing learning step, provided in **Theorem E.8** and **Theorem E.9**, are deferred to **Appendix E**. As the proof sketch, we first assume small i.i.d. noise for the realization of $\phi_k(\theta)$ and its gradient (**Assumption E.4**), then get the variance of updating step (**Lemma E.5** and **Lemma E.6**) followed by stochastic convergence proof. We also discuss the stochastic convergence rate under a fix learning step setup.

## 3.4 REGULATING SPECTRAL RADIUS AS FAST PRE-PROCESSING

When an unstable system is controlled by a random policy and initialized by an arbitrary condition, the internal states/outputs will most likely grow unboundedly due to the system instability. In many of the RL scenarios, agent stability is a required but usually neglected property for optimal or near-optimal solutions. For a discrete linear system, the spectral radius (largest absolute value of system eigenvalues) being strictly less than 1 is a sufficient condition for exponential stability. For non-linear systems without an explicit form of state transition matrix, we expect the next time step state norm does not deviate too much from the current step. It is computationally costly to optimize the value function from initial policy. To speed up the optimization, we propose a fast pre-processing method by finding a policy close to the stable zone. The algorithm is shown in Algorithm 1. In the pre-process, we neglect the value function and only regulate the spectral radius of the system dynamics estimated by power iteration.

---

**Algorithm 1** Regulating system spectral norm by power iteration

**Input:** system state transition function $f$, finite time step $T$, batch size $b$,
Initialize policy parameter $\theta$
**for** $l = 1$ **to** $N$ **do**
    Sample $\{x_0^1 ... x_0^i ... x_0^b\}$
    $Loss \leftarrow 0$
    **for** $t = 0$ **to** $T$ **do**
        **for** $i = 1$ **to** $b$ **parallel do**
            $x_t^{i+1} \leftarrow f(x_t^i, \theta)$
            $Loss \leftarrow Loss + \max(\frac{\|x_T^i\|}{\|x_{T-1}^i\|} - 1, 0)$
        **end for**
    **end for**
    $Loss \leftarrow Loss/b$
    Update parameters: $\theta \leftarrow \theta - \eta \nabla_\theta Loss$
**end for**
**Output:** policy parameter $\theta$

---

---

**Algorithm 2** Policy Gradient for LQR

---

**Input:** system state transition matrix $A$ and $B$, quadratic cost matrix $Q$ and $R$, finite time step $T$, batch size $b$, discount factor $\gamma$
Initialize policy matrix $K_0$
**for** $l = 1$ **to** $N$ **do**
    Sample $\{x_0^1...x_0^i...x_0^b\}$
    $V \leftarrow 0$
    **for** $i = 1$ **to** $b$ **parallel do**
        **for** $t = 0$ **to** $T$ **do**
            $u_t^i \leftarrow K x_t^i$
            $x_t^{i+1} \leftarrow A x_t^i + B u_t^i$
            $V \leftarrow V + \gamma^t({x_t^i}^\top Q x_t^i + {u_t^i}^\top R u_t^i)$
        **end for**
    **end for**
    $V \leftarrow V/b$
    Update parameters: $K \leftarrow K - \eta \nabla_K V$
**end for**
**Output:** policy matrix $K$

---

## 4 EXPERIMENTS

In this section, we use LQR as illustrative examples to demonstrate the efficacy of our methods. We also apply these methods to general unstable RL applications in continuous control and defer the results to Appendix D. Note that in our implementation of policy gradient methods, we assume the gradient from penalty towards model parameters is directly accessible, despite of the implicit environment dynamics to the optimization process. With the focus of the paper being on the controller decision for the system, our implementation trains a deterministic actor model instead of a full actor-critic model. We bypass the critic modeling with a directly approximated critic function, but our methods could be extended to a more general RL algorithm such as actor-critic.

### 4.1 LQR

The LQR system matrices $A$, $B$ and positive-definite cost matrices $Q$, $R$ are randomly generated. To regulate the "instability" of the autonomous system, the spectral radius of $A$ can be manipulated by multiplying the original random matrix with a ratio between the desired spectral radius and that of the original random matrix. A vanilla LQR policy gradient method is shown in Algorithm 2. In each batch, we run 10 fixed time steps from 300 random initial states. For each experiment, we run 3 random seeds for reproducibility. We use Pytorch (Paszke et al., 2019) as the optimization platform and carry all the simulations on single CPU since the tasks are relatively light-duty.

Figure 1a shows the vulnerability of optimization for unstable systems with large learning rates. For $\rho_{max}(A) = 0.5$ and 1, the model is trainable across all the learning rates, where larger learning rates lead to smaller losses after 100 epochs. For $\rho_{max}(A) = 1.5$, the largest 2 learning rates could crash the optimization and only the smallest learning rate works for $\rho_{max}(A) = 2$. The observation is consistent with Lemma 3.7's claim that the optimization step size should be bounded for training stability. However, a small learning rate could result in slow convergence. Therefore, the major challenge for vanilla policy gradient method on unstable RL problem is how to find an optimal learning rate without crashing the optimization. In our experiments, we test different learning rates by log intervals such as $\{1e\text{-}1, 1e\text{-}2, ... \}$ and select the largest one without breaking the optimization (if any 1 of 3 random experiments fails, we consider that the optimization failed and then use a smaller learning rate).

In Figure 1b we compare the vanilla gradient method and its variants with our proposed logarithmic mapping. The subplots are normalized cost difference towards optimal, normalized policy difference to the optimal and estimated spectral norm by power iteration, respectively. Note that the y-axis of the cost figure is log scale. The optimal policy $K^\star$ is,

$$K^\star = (B^\top P B + R)^{-1} B^\top P A,$$

where $P$ is the solution to the infinite horizon algebraic Riccati equation,

$$P = A^\top (P - PB(B^\top PB + R)^{-1}B^\top P)A + Q.$$

Because the sampled *cost* is approximated by rolling out from many random initial states, it could be smaller than theoretical *cost*$^\star$. Therefore, the cost difference plot could disappear in the log scale when the solution is very close to optimal policy. Besides, we only draw the seed variance above the value mean in the log scale for clear visualization.

When optimizing the model with vanilla sum loss function from random initialization without any pre-processing, the system starts from an extremely unstable condition with large cost. From Lemma 3.7, a small learning rate is needed when the system is unstable with control policy parameters. In practice, $1e$-14 is the maximum learning rate to accommodate the instability with $1e$-13 crashing the optimization. If we run an efficient policy pre-process with a total of 300 episodes, we could use $1e$-8 as learning rate. During the optimization, the controller gradually stabilizes and the fixed learning rate is then too small for further parameter update. The policy and corresponding cost move very slowly to optimal and almost stagnate at certain point, and optimization is impractical. We also tested other discount factor such as $\gamma = 0.5$ to shrink the cost and allow a larger learning rate $1e$-7, but this method also fails because the small $\gamma$ neglects the long term effect of the dynamics with spectral radius reaching 2. Combining our proposed pre-process and log mapped loss function, we are able to use $1e$-2 as learning rate and both cost and policy parameters approach optimal quickly.

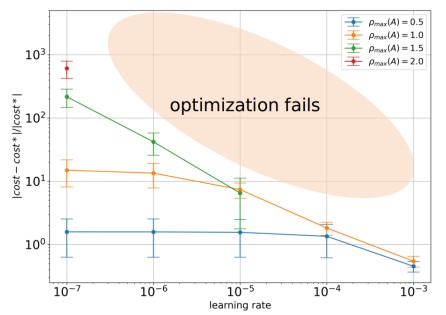

(a) Vanilla PG LQR loss difference to optimal after 100 epochs under different learning rates and spectral radius (missing point for $\rho_{max} = 1.5, 2$ means the cost goes to NaN when optimization crashes)

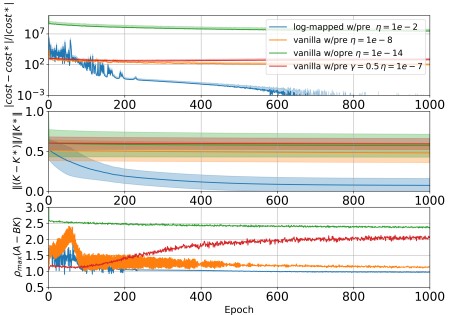

(b) LQR loss difference to optimal: vanilla loss vs log mapping, $\rho_{max}(A) = 5$. Combining logarithmic mapping and pre-processing, our method reaches near-optimal solution quickly and other methods are likely to diverge.

Figure 1: LQR experiments

Other than $\rho_{max}(A) = 5$ case, we further conducted experiments with $\rho_{max}(A) = 2$ and $\rho_{max}(A) = 10$ and defer them to Appendix C.

## 5 CONCLUSION AND FUTURE WORK

This paper focuses on the gradient-based optimization for a special branch of RL problems. Due to the unstable nature of the system, small deviation leads to exponentially growing effects on the state evolution trajectory and the reward/cost function, which raises issues for gradient-based optimizations. We proposed two methods to alleviate the effect of instability and their effectiveness is validated from both theoretical and experimental points of view.

Regarding the future work, proving the global convergence in our finite horizon setup is needed to extend (Fazel et al., 2018)'s work to unstable cases. In this paper, we examine the experiments only on model-free RL where the reward functions are directly approximated. Our methods has the potential to be applied to other RL algorithms such as actor-critic. In addition, for model-based RL, coupling stability regularization into the dynamics modeling could be another path to tackle the unstable RL problems.

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
