# OpenReview forum: "On the Fast Convergence of Unstable Reinforcement Learning Problems"
_ICLR.cc/2023/Conference — Submitted to ICLR 2023_

### Official Review · Reviewer_1dw6 · 2022-10-21

**Confidence:** 1
**Correctness:** 3
**Technical Novelty And Significance:** 3
**Empirical Novelty And Significance:** 3
**Recommendation:** 5

**Clarity, Quality, Novelty And Reproducibility:**

The connection with the LQR controller could be better clarified for a less involved audience. The paper motivates the use of LQR controllers, but then seems to prove theorems that are more general.

I also did not understand how the theoretical insights (for example the log-transformation) are translated into Algorithm 1 and 2. Alg.2 seems to be gradient descend on standard LQR, whereas Alg.1 seems to do something similar for general differentiable transition functions some additional spectral norm regularization. However, it is not clear to me how this is related to the convergent guarantees of the theorems above.

**Strength And Weaknesses:**

I have to admit that I did not understand this paper. In particular, I cannot say how realistic Assumption 3.2 is, and how Theorems 3.14 and 3.15 proof stability. However, the experimental results look good, even though I find it hard to say how general they are.

**Summary Of The Paper:**

The paper analyzes the convergence of policy gradients on I/O unstable problems, in particular unstable LQR problems. Under a variety of assumptions, the authors claim to proof an bound on the learning rate, related to the largest spectral eigenvalue of the dynamics matrix of LQR. Experiments show that a log mapping of the objective stabilizes gradient descend on LQR's policy matrix K.

**Summary Of The Review:**

I admit that I was not able to follow the paper. I am uncertain whether this is the fault of the paper or mine. While I recommend to reject the paper, I am happy to change my mind if more qualified reviewers think the paper deserves publication.

**Post-rebuttal**

I thank the authors for their attempt to explain their paper, but I still feel that I do not understand the paper enough to give a certain recommendation. Due to this uncertainty I retain my previous recommendation to reject the paper.

---

> ### Author Response · Authors · 2022-11-12
> **Author rebuttal**
>
> Dear Reviewer 1dw6:
>
> Thank you for your frank and valuable comments, we would like to address your concerns one by one in the following. We summarize the narrative thread for our theoretical analysis in the response below and we hope it could help you fully understand our paper. We sincerely wish you can discuss with us and allow us a chance to explain our paper, as well as clarify the further confusion around it.
>
> **$\bullet$ Assumption 3.2**: We introduce assumption 3.2 to facilitate the theoretical analysis. The intuition of exponential basis function decomposition was inspired from the LQR example, as illustrated in Remark 3.3. In such setup, we can bound the exponential power instead of the step cost function, allowing the latter to grow exponentially in time. However, we do require the cost function to be positive, otherwise the logarithm operation is not legit.
>
> **$\bullet$ LQR introduction**:
> Thank you for the suggestion. We use the LQR example just as introductory examples to illustrate the unstable issue in unstable optimization, as it is the most studied example in control community. To facilitate the readers from other backgrounds, we use sec 2.2 to describe the LQR setup (eqn 1,2,3,4) and Alg 2 for the policy gradient method for LQR. We also discuss the analytical LQR solution in sec 4.1 and compare it with our numerical experiments. In terms of the connection between LQR and our general setup, following your suggestions, we explicitly add the connection in Remark 3.3 and mark the text in red. Finite horizon LQR example is a special case of our formulation above, where the control matrix $K$ and cost function $C_{K,T}$ corresponds to the policy parameter $\theta$ and cost function $V_{T}(\theta)$, The LQR step cost $x_t^\top Qx_t + u_t^\top Ru_t$ can be rewritten as $_0^\top {(A-BK)^t}^\top (Q+K^\top R K) (A-BK)^t x_0 $
>
>
> **$\bullet$ Theorem 3.14 and 3.15**: These theorems are not proof of stability. They show the convergence rate bound under a “safe” update step size, under such step size the cost function is guaranteed to decrease during optimization. We would like to draw a storyline for the theoretical work to help clarify. Due to the unstable nature of the problem, the policy optimization is very sensitive to parameter change. Lemma 3.7 describes the sufficient condition of the update step size for the cost function to monotonically decrease. Theorem 3.9 and 3.10 converts the step requirement with cost function under our assumption 3.2, 3.4 and 3.5. As one of the major results, Theorem 3.12 shows the convergence bound under vanilla policy gradient method if we choose the fixed step size that satisfies the requirement. Theorem 3.14 and 3.15 are the counterpart of 3.10, 3.12 under logarithmic mapping, where we can achieve a faster convergence bound with fixed learning rate that also satisfies Lemma 3.7.
>
> **$\bullet$ Algorithm 1 and 2**: Alg 1 is an intuitive pre-processing to help find a better initialization for the main optimization.  Alg 2 is for standard/vanilla gradient descent for LQR problem. Our proposed method only needs to replace the sampled cost function with a logarithmic mapped one, as illustrated in sec 3.2.1 (eqn 11 and 12).

---

### Official Review · Reviewer_tyRg · 2022-10-24

**Confidence:** 3
**Correctness:** 3
**Technical Novelty And Significance:** 2
**Empirical Novelty And Significance:** 2
**Recommendation:** 6

**Clarity, Quality, Novelty And Reproducibility:**

The paper seems to include novel results.
The presentation needs to be improved as mentioned in the previous comments.

**Strength And Weaknesses:**

Overall, the paper seems to contain interesting results.
Some comments to improve the paper are summarized below:
1) For an MDP with finite and discrete state and action spaces, the stable MDP is not clear.
Only for MDPs with continuous state and action spaces, that are main focus of the paper, stability is meaningful.
In abstract and introduciton, the system is not clearly indicated, and hence, it is very confuzing at the beginning on what stability means for MDPs.
Therefore, it would be better to indicate that the system is an MDP with continuous spaces at the beginning.
2) In assumption 3.2, it is not clear what is the basis function. In the explanation after assumption 3.2., it says that || A-BK||^2 corresponds to \phi.
However, || A-BK||^2 is scalar number, while \phi needs to span the function space. This discussion seems to be unreasonable.
3) The convergene results in Theorem 3.9 and Theorem 3.10 are not clear. It is not clear why the inequality leads to convergence. Some improvement of presentation may be needed.
4) It would be better to indicate earlier in section 3.2.2. that the results are all for deterministic settings.

**Summary Of The Paper:**

This paper focuses on the gradient-based optimization for a special branch of RL problems. Due to the unstable nature of the system, small deviation leads to exponentially growing effects on the state
evolution trajectory and the reward/cost function, which raises issues for gradient-based optimizations. The authors proposed two methods to alleviate the effect of instability and their effectiveness is validated from
both theoretical and experimental points of view.

**Summary Of The Review:**

This paper focuses on the gradient-based optimization for a special branch of RL problems.
Overall, the paper seems to contain interesting results.
The presentations need to be improved as mentioned in the previous comments.

---

> ### Author Response · Authors · 2022-11-12
> **Author rebuttal**
>
> Dear Reviewer tyRg:
>
> Thank you for the positive feedback and pointing out your concerns, they are of great value to us, and we are very glad to clarify them in the following. As a short summary of the detailed paragraph below, 1. we improve the presentation in the new manuscript following your suggestions. 2. we clarified your confusion one by one in the following. We hope our answers address all your concerns. Please do not hesitate to contact us if you need any additional clarifications. We will do our best to resolve all your concerns.
>
> **$\bullet$ Continuous MDP and stability**: Thank you for the suggestion, we added “continuous state” description when introducing MDP formulation in sec 3.2.1. Regarding the stability, as mentioned in our paper (footnote 3, section 3.1, appendix A), we denote stability in this paper as input-output (IO) stability instead of input-state (IS) stability, where a small perturbation of input action/policy might lead to a dramatic change in output (reward, cost).
>
> **$\bullet$ Assumption 3.2**: Analogy between $||A-BK||^2$ and $\sigma$: Thank you for pointing out the confusion. By assumption 3.2, we aim to disassemble the target function into a combination of basis functions to facilitate convergence analysis. As the application to the LQR problem, this is a special case where we only need 1 basis scalar function $\sigma(K)=||A-BK||^2$. Here $\sigma(K)$  is dependent on the control matrix parameter $K$ ($K$ corresponds to policy parameter $\theta$ in our later derivation). We introduce this analogy to help readers better understand our approach through the LQR example.
>
> **$\bullet$ Convergence results in Theorem 3.9 and Theorem 3.10**. Thank you for raising the confusion. Theorem 3.9 and Theorem 3.10 are intermediate steps leading to the final result Theorem 3.12. In Lemma 3.7, we derive the update step size requirement to guarantee monotonic decrease of cost function and avoid overshooting (or potential crash in optimization). Theorem 3.9 and Theorem 3.10 converts the step requirement with cost function under our assumption 3.2, 3.4 and 3.5. Then Theorem 3.12 gives the convergence rate bound for vanilla gradient method if the fixed step size satisfies the requirement. We also derive the counterpart theorem for our proposed method in Theorem 3.12. The convergence bound is much faster under logarithmic mapping.
>
> **$\bullet$ deterministic settings**. Thank you for the suggestion. Most of the derivation results are from deterministic optimization (sec 3.2) and we also include further discussion on stochastic cases in sec 3.3. Following your suggestion, we have explicitly indicated in the analysis roadmap (sec 3.2) that sec 3.2 is for deterministic cases, and sec. 3.3 is for stochastic cases with red text in the new manuscript.

---

> > ### Comment · Reviewer_tyRg · 2022-12-14
> > **Response to author's response**
> >
> > I thank the authors for the clarification. The authors addressed my previous comments well. I would like to keep my previous score.

---

### Official Review · Reviewer_xiDA · 2022-10-24

**Confidence:** 4
**Clarity, Quality, Novelty And Reproducibility:** See above.
**Correctness:** 4
**Technical Novelty And Significance:** 3
**Empirical Novelty And Significance:** 3
**Recommendation:** 3

**Strength And Weaknesses:**

I like the general idea of the paper; it has an understandable approach and puts forward an interesting and to some extent important problem. The writing in most of the manuscript is acceptable, but some improvements help. The technicalities seems based on strong assumptions though.

The deterministic nature of the dynamics restricts applicability of the approach to data-driven problems. Also importantly, in case of randomly or adversarially disturbed dynamics, the stabilization needs to be different since other than the initial state, the innovation can cause instability. Although the authors claim that the method is applicable to disturbed dynamics, the analysis does not seem so. Further, the existing literature of learning-based stabilization is not covered (enough). The setting and analysis of the GD does not seem innovative and seems a little artificial. The enumeration looks strange. In some places the language is not articulate enough, e.g., in Lemma 3.7.

The technical assumptions, e.g., 3.3 and 3.4, are a little restrictive and seem consequential. So, I think further justifications and intuitive arguments are needed. Importance of the main results Thm 3.9 and 3.10 are not clear, and it is also unclear what technical difficulties need to be addressed to obtain them. Finally, considering the step size upper bounds, applicability of the algorithms to only small time horizons renders the framework limited in the sense that the step size will be so small that the method will have no superiority to other learning based stabilization approaches, even if it is somehow applicable.

The edits the authors provided in the resubmitted version do not seem adequate. I suggest the authors to do further rewrites and to go deeper for proving interesting results.

**Summary Of The Paper:**

The paper considers gradient descent algorithms for computing a parameter of the optimal policy according to a possibly unstable trajectory of a Markov decision process. It is shown that under technically strong assumptions such as continuity, convexity, and smoothness, the rates of convergence will be linear, assuming that the horizon is short enough.

**Summary Of The Review:**

I prefer to give the authors a chance to see how they can convince the reviewers that their paper is interesting, important, and innovative.

--------------------------------
post rebuttal: this reviewer carefully read the rebuttal and updated the review.

---

> ### Author Response · Authors · 2022-11-12
> **Author rebuttal**
>
> Dear reviewer xiDA:
>
> Thank you for pointing out your concerns about our paper, they are of great value to us, and we are very glad to clarify them in the following. As a short summary of the detailed paragraph below, 1. Our assumptions are strong but common in optimization literature, and it does not interfere with our contribution to unstable RL problems 2. We have improved the enumeration following your suggestions and update it in the new manuscript in red. We hope our answers address all your concerns. Please do not hesitate to contact us if you need any additional clarifications. We will do our best to resolve all your concerns.
>
> **$\bullet$ technical assumptions**: We agree that our assumptions are relatively strong. The Lipschitz continuity and strong convexity assumption are commonly used in optimization papers Ref[C]. Reinforcement learning optimization is in general nonconvex. However, convex assumptions or alternative convex surrogate functions (Ref[B]) are often used to facilitate the derivation. In our setup, we analyze the optimization convergence towards the stationary point around the local convexity, therefore the assumptions are legit in our scenario. The further justification of our assumption is in Remark 3.6. Besides, all these assumptions are used to facilitate the theoretical analysis for unstable RL problems and our innovation mainly lies on the latter instead of the optimization problem itself.
>
> **$\bullet$ Deterministic nature of the dynamics**: Thank you for pointing it out. We agree that all our experiments are deterministic. The purpose of these experiments is to demonstrate the convergence issue for unstable problems and show our method could address that. We choose these deterministic control dynamics as examples to help readers better understand the instability issue and link it to our approach. However, in the theoretical analysis, our cost function is generally applicable to both deterministic and stochastic dynamics, as the target function can be seen as the expectation of future costs and estimated by sampling in practice. The method is not limited to deterministic problems only.
>
> **$\bullet$ Lemma 3.7 and Thm 3.9 and 3.10**: We change the enumeration for the corresponding theorems for clarification and mark the modification in red in the new manuscript. For Lemma 3.7, we cite the proposition 3.4 in Ref [A] as the inspiration source for readers’ convenience. Lemma 3.7 describes the necessary condition on step size for the monotonically decreasing during function update. In Thm 3.9 and 3.10, we customize the step size requirement of Lemma 3.7 for $V_T(\theta)$ satisfying Assumption 3.2 and 3.4, to guarantee the monotonically decreasing for vanilla gradient descent. The main result for vanilla gradient descent is in Thm 3.12: in order to guarantee the monotonic decrease and avoid potential crash in optimization, the largest update step should be limited and therefore the convergence rate is also bounded. As for our proposed method, the counterpart of Thm 3.12 is Thm 3.15. With logarithmic mapping, we can use a larger update step and achieve faster convergence.
>
> **$\bullet$ applicability of the algorithms to only small time horizons**: thank you for pointing out the time horizon issue, and you might misunderstand our paper. In sec 3.2.1, we formulate the problem into finite horizon setup because the infinite horizon problem requires bounded step cost function, otherwise the to-go cost function could be infinite and intractable. Indeed, finite horizon setup is the only way to deal with a cost function exponentially growing with time (see eqn (7)). However, this does not mean it is limited to a small horizon, we allow the time step T to be large enough (as long as it does not go to infinite). In fact, even the infinite horizon problem is always estimated by finite horizon implementation. You mentioned “Considering the step size upper bounds, applicability of the algorithms to only small time horizons renders the framework limited.”, we guess you might mistake the optimization update step with the time step. We do limit the update step size, but do not require the discrete time step to be small.
>
> Ref [A]: Dimitri Bertsekas and John N Tsitsiklis. Neuro-dynamic programming. Athena Scientific, 1996.
>
> Ref[B]: Yu, Ming and Yang, Zhuoran and Kolar, Mladen and Wang, Zhaoran Convergent policy optimization for safe reinforcement learning. NeurIPS 2019
>
> Ref[C] Lam Nguyen, Phuong Ha Nguyen, Marten Dijk, Peter Richtarik, Katya Scheinberg, and Martin Takac. Sgd and hogwild! convergence without the bounded gradients assumption. ICML 2018

---

### Author Response · Authors · 2022-11-14
**General reply to all the reviewers:**

Dear reviewers:

We would like to thank all the reviewers for their thoughtful comments. We appreciate the positive assessment and also value constructive suggestions from all the reviewers.

For the individual concerns/questions raised by the reviewers, we replied one by one in the following. We hope our answers address all your concerns. Please do not hesitate to contact us if you need any additional clarifications. We will do our best to resolve all your concerns. When all your concerns are resolved, we sincerely hope that you could increase your score, as our contribution is the first attempt to investigate the convergence issues of policy gradient methods on unstable systems and propose practical solutions to alleviate the issues.

Authors

---

### Decision · Program_Chairs · 2023-01-20

**Decision:**

Reject

**Justification For Why Not Higher Score:**

There are a number of weaknesses presented in the paper, and the authors failed to convince the reviewers that the paper contains significant contributions to be published in ICLR.

**Justification For Why Not Lower Score:**

N/A

**Metareview: Summary, Strengths And Weaknesses:**

The paper studies the policy gradient method of unstable dynamic systems. The stability considered in this paper is the so-called "input-output" stability (I/O), where a function of the system state is always bounded by a function of the initial state and time. The paper considers gradient descent algorithms for computing a parameter of the optimal policy according to a possibly unstable trajectory of a Markov decision process. It is shown that under technically strong assumptions such as continuity, convexity, and smoothness, the convergence rates will be linear, assuming that the horizon is short enough.

Strengths:
- it has an understandable approach and puts forward an interesting and, to some extent, important problem

Weakness:
- contains a series of strong technical conditions, which might not be realistic
- presentation is not clear enough: it is hard to get a clear understanding of why the instability conditions could lead to difficulties in the algorithm and analysis. There is a lack of discussion on the input/output stability (it looks like ISS is a special case of I/O stability, and these functions $\beta$, $\gamma$ are also not clear, can they be arbitrary functions?). The paper also contains a number of typos that make it hard to understand what is going on.

The metareviewer suggests the authors to address the concerns from the reviewers and improve the paper accordingly.